# The Capability of O-Acetyl-ADP-Ribose, an Epigenetic Metabolic Small Molecule, on Promoting the Further Spreading of Sir3 along the Telomeric Chromatin

**DOI:** 10.3390/genes10080577

**Published:** 2019-07-30

**Authors:** Shu-Yun Tung, Sue-Hong Wang, Sue-Ping Lee, Shu-Ping Tsai, Kuan-Chung Su, Hsiao-Hsuian Shen, Jia-Yang Hong, Ming-Shiun Tsai, Gunn-Guang Liou

**Affiliations:** 1Institute of Molecular Biology, Academia Sinica, Taipei 115, Taiwan; 2Department of Biomedical Sciences, Chung Shan Medical University, Taichung 402, Taiwan; 3Department of Medical Research, Chung Shan Medical University Hospital, Taichung 402, Taiwan; 4Institute of Molecular and Genomic Medicine, National Health Research Institutes, Miaoli 350, Taiwan; 5Department of Food Science and Biotechnology, Da-Yeh University, Changhua 515, Taiwan; 6Institute of Biological Chemistry, Academia Sinica, Taipei 115, Taiwan; 7Guang EM Laboratory, New Taipei 242, Taiwan

**Keywords:** AAR (O-acetyl-ADP-ribose), epigenetic, silent heterochromatin, Sir2, Sir3, small molecule, telomere

## Abstract

O-acetyl-ADP-ribose (AAR) is a metabolic small molecule relevant in epigenetics that is generated by NAD-dependent histone deacetylases, such as Sir2. The formation of silent heterochromatin in yeast requires histone deacetylation by Sir2, structural rearrangement of SIR complexes, spreading of SIR complexes along the chromatin, and additional maturation processing. AAR affects the interactions of the SIR-nucleosome in vitro and enhances the chromatin epigenetic silencing effect in vivo. In this study, using isothermal titration calorimetry (ITC) and dot blotting methods, we showed the direct interaction of AAR with Sir3. Furthermore, through chromatin immunoprecipitation (ChIP)-on-chip and chromatin affinity purification (ChAP)-on chip assays, we discovered that AAR is capable of increasing the extended spreading of Sir3 along telomeres, but not Sir2. In addition, the findings of a quantitative real-time polymerase chain reaction (qRT-PCR) and examinations of an in vitro assembly system of SIR-nucleosome heterochromatin filament were consistent with these results. This study provides evidence indicating another important effect of AAR in vivo. AAR may play a specific modulating role in the formation of silent SIR-nucleosome heterochromatin in yeast.

## 1. Introduction

Epigenetic inheritance of chromatin is a non-DNA sequence-dependent phenomenon. Chromatin is packaged from DNA together with other associated proteins and molecules. It is organized in euchromatin and heterochromatin states [1,2,3]. Transcriptionally active portions of DNA are termed euchromatin, whereas transcriptionally less-active condensed chromosome regions are termed heterochromatin. The established silent heterochromatin regions of the budding yeast, *Saccharomyces cerevisiae*, a eukaryotic microorganism, are silent mating type loci (*HML* and *HMR*), telomeric regions and ribosomal DNA (rDNA) tandem repeats [2,4,5,6].

The silent information regulator (Sir) proteins, Sir2, Sir3, and Sir4, associate with each other as a SIR complex [7,8,9,10]. SIR complexes are required for the establishment and maintenance of silent heterochromatin domains at telomeres and mating type loci [11,12,13,14]. O-acetyl-ADP-ribose (AAR), an epigenetic metabolic small molecule, is generated by the NAD-dependent Sir2/Class III histone deacetylase (HDAC) protein [15,16,17,18,19]. AAR associates with silent heterochromatin domains, shows a similar pattern to the genome-wide localization of Sir2 [20], and enhances yeast chromatin epigenetic gene silencing in vivo [21]. It induces a change in both the conformation and stoichiometry of the SIR complex [22] and increases the binding affinity of Sir3 to nucleosomes in vitro [23]. In addition, an in vitro assembly system of SIR-nucleosome pre-heterochromatin filaments that has been established. All purified yeast nucleosomes, Sir2, Sir3, Sir4, and AAR, are required for this assembly, and the formation of these filaments is modulated by AAR. Overall, these filaments exhibit requirements that closely mirror those observed for the formation of silent heterochromatin in vivo [24,25]. Furthermore, it has been suggested that at least one of the Sir proteins is an AAR-binding partner [22]. AAR is known to directly interact with Sir2 [20]. Notably, an AAA ATPase-like domain in the Sir3 C-terminal region may interact with ATP [26], and AAR exhibits a structural similarity to ATP. Hence, Sir3 may be another AAR-binding partner.

Based on a number of genetic and biochemical studies, several models for silent heterochromatin formation have been proposed [4,20,22,27,28]. The current model of silent heterochromatin formation is based on the spreading of SIR complexes along the chromatin fiber. On the initial establishment processes, Sir2 deacetylates histones as a hub to associate with Sir3 and Sir4. This leads to repeated recruitment cycles of SIR complexes along the adjacent chromatin region to form an extended silent heterochromatin domain. Additional maturation processes should stabilize and maintain the silent heterochromatin state through more epigenetic modifications, the binding of small molecules, and/or protein conformational changes [27,28,29,30]. However, the detailed regulatory mechanisms involved in this process remain more investigations.

In this study, we investigated in detail the mechanism of silent heterochromatin formation in yeast. We showed the interaction of AAR with Sir3 and discovered that AAR modulates the spreading of Sir3 along the telomeres. Our results reveal another important function of AAR on the formation of silent SIR-nucleosome heterochromatin in vivo.

## 2. Materials and Methods 

### 2.1. Yeast Strains

The yeast strains used in this study are listed in Appendix A. Epitope-tagged strains were constructed using a PCR-based gene-targeting method, as previously described [24,31].

### 2.2. Purification of Proteins and AAR

The purification of Sir3-CBP, Sir3N_(1-214 aa)_-CBP, Sir3C_(215-978 aa)_-CBP, native yeast nucleosomes, and AAR was based on previously described methods [20,22,24]. Through high performance liquid chromatography (HPLC), the fraction of the AAR peak was collected, lyophilized, and identified through MALDI-TOF mass spectrometry. ^32^P-AAR was prepared by deacetylation of an acetylated histone H4 peptide using Hst2 and ^32^P-NAD.

### 2.3. Dot Blotting

Sir3 and Sir3C were individually spotted on a polyvinylidene difluoride (PVDF) membrane with a serial dilution of 50–200 μmole and incubated with ^32^P-AAR (or ^32^P-NAD) at 4 °C overnight. The membrane was washed twice with 20 mM HEPES-KOH pH 8.0, 300 mM KCl, 1 mM MgCl_2_, 0.05% NP40, and washed twice with 20 mM HEPES-KOH pH8.0, 100 mM KCl, 1 mM MgCl_2_, 0.025% NP40 at room temperature for 30 min. Then, the membrane was analyzed using autoradiography.

### 2.4. Isothermal Titration Calorimetry (ITC)

The isothermal titration calorimetry (ITC) measurements were performed on a VP-ITC system (MicroCal-GE Healthcare, Piscataway, NJ), using the VPViewer software for data acquisition and instrument control. All proteins were extensively dialyzed with 20 mM HEPES-KOH pH8.0, 300 mM KCl, 1 mM MgCl_2_. All samples and buffers were filtered and degassed before use. In the experiment presented in Figure 1, the concentration of AAR in the injection syringe was 50 μM, and the concentration of Sir3 (or Sir3C) in the reaction cell was 5 μM. In a typical experiment, after an initial 1.5 μL injection, 29 aliquots of 10 μL were titrated at 4 min intervals from the syringe into the 1.4 mL sample cell, maintained at 20 °C, and stirred at a constant rate of 270 rpm to ensure rapid mixing. The four titration curves of binding isotherm data were analyzed using the MicroCal Origin software package, assuming one set of sites to obtain the binding constant (K), reaction stoichiometry (N), enthalpy (ΔH), and entropy (ΔS) with their standard errors for fitting to the data. 

### 2.5. Chromatin Immunoprecipitation on Chip (ChIP-on-chip), and Chromatin Affinity-Precipitation on Chip (ChAP-on-chip)

The chromatin immunoprecipitation on chip (ChIP-on-chip) and chromatin affinity-precipitation on chip (ChAP-on-chip) assays were performed as previously described [20], with some modifications. Briefly, precipitated chromatin DNA fragments of Sir3-Myc, Sir2, and AAR were obtained through association with a Myc antibody (Merck Millipore, Burlington, MA), a Sir2 antibody (Merck Millipore, Burlington, MA), and purified AAR, respectively. According to the manufacturer’s protocol for NimbleGen (Roche NimbleGen, Madison, WI), the samples were subjected to post-immunoprecipitation, ligation-mediated PCR, Cy3 (or Cy5) labeling, hybridization with a tiling array chip (*Saccharomyces cerevisiae* whole genome, Roche NimbleGen, Madison, WI), and chip scanning. Data analyses were conducted with the NimbleScan program and SignalMap software. The data were deposited in National Center for Biotechnology Information Gene Expression Omnibus (NCBI GEO) (accession #GSE65672).

### 2.6. Quantitative Real Time Polymerase Chain Reaction (qRT-PCR)

The quantitative real time PCR (qRT-PCR) assays were performed on a Bio-Rad CFX Real-Time PCR detection system using the iQ SYBR Green Supermix (Bio-Rad, Hercules, CA) in triplicate at least twice. The primers used are listed in Appendix A. The concentration of each ChIP (or ChAP) DNA sample was diluted to 1/20 for actin compared with the concentration for the examined genes. Using the equipment accompanying the CFX manager software, the threshold cycle (Ct) values were acquired. The relative ratios of the expressional levels were quantified using the comparative 2^-ΔΔCt^ method normalized against the levels of actin and compared with their respective input DNA.

### 2.7. Electron Microscopy (EM)

The in vitro reactions of the SIR-nucleosome were performed as previously described [20,24] with some modifications. Briefly, the Sir3 and Sir3N-Sir3C mixture (~0.8, ~0.2, ~0.6 mg/mL, respectively) were each incubated with nucleosomes (~0.2 nM), with or without AAR (~8 mM), in a 10 μL reaction volume with 50 mM HEPES-KOH pH 7.6, 300 mM KCl, 4 mM MgCl_2_, at room temperature for 2–4 h and then overnight with rocking at 4 °C. Samples were then examined by electron microscopy (EM) as previously described [22].

## 3. Results and Discussion

### 3.1. Interaction of AAR with Sir3

Firstly, we used a dot blotting assay to rapidly determine the possible binding of AAR to a Sir3 protein that was immobilized on a PVDF membrane. We blotted increasing amounts of Sir3 (50–200 μmole) onto membranes, which were then incubated with ^32^P labelled AAR. As shown in Figure 1a, we detected AAR binding to Sir3 on the membrane. In contrast, we did not observe binding to Sir3C, which contains the AAA ATPase-like domain of Sir3. This implies that Sir3C was not sufficient for AAR binding and suggests that this fragment of Sir3 requires other region(s) to form an entire binding pocket for AAR or may not even be the AAR binding region. 

We used the isothermal titration calorimetry (ITC) assay to measure the interaction of AAR with Sir3 under equilibrium conditions. In this assay, AAR (50 μM) was injected into a reaction cell containing Sir3 (5 μM). As shown in Figure 1b,c, the interaction of AAR with Sir3 exhibited an apparent K of ~3.3 × 10^7^ M^−1^, exothermic properties (ΔH = -2070 kcal/M), and a large negative entropy value (ΔS = -7025 cal/M/deg), suggesting that binding may be accompanied by significant structural rearrangements. We also plotted the integrated heats of binding as a function of the molar ratio of AAR to Sir3. The stoichiometry of AAR binding to Sir3, calculated from the ITC binding data, was 0.81. This may indicate a 1:1 molar ratio of AAR to Sir3. The results of the ITC assays were consistent with those obtained from the dot blotting experiment. Although interaction of AAR with Sir3 was observed (Figure 1c), there was no interaction noted between AAR and Sir3C (Figure 1d).

### 3.2. Occupancies of AAR, Sir2 and Sir3 on Telomeres

AAR directly interacts with Sir2 [20] and Sir3 (as described above). In addition, AAR is able to enhance the chromatin epigenetic silencing effect [21] that prompted us to investigate whether AAR may also promote an extended distribution of Sir proteins along chromatin.

We used chromatin immunoprecipitation (ChIP)-on-chip and chromatin affinity purification (ChAP)-on-chip assays to map the distributions of Sir2, Sir3, and AAR on telomeres. As expected, in general, overlapping distributions of Sir2, Sir3, and AAR, extending to telomeric DNA regions for all chromosomes, were observed in wild type cells (Figure 2).

AAR is metabolized to AMP by Ysa1, and the amount of AAR increases by approximately 50% in cells of the *ysa1*-deletion strain compared with wild-type cells [32]. Increased cellular AAR enhances the chromatin silencing effect [21]. Theoretically, increased cellular AAR may also result in the extensive spreading of AAR and AAR-associated Sir proteins along the chromatin to form a longer silent heterochromatin. To test this hypothesis, we compared the occupancies of Sir2, Sir3, and AAR in the *ysa1*-deletion strain cells and wild-type strain cells.

Interestingly, using the same ChIP-on-chip and ChAP-on-chip approaches, we found that both Sir3 and AAR spread further along chromosomes in the *ysa1*-deletion cells versus wild-type cells. In contrast, the distribution patterns of Sir2 were similar between the wild-type and *ysa1*-deletion cells. In addition, Sir3 and AAR were associated with a region that extended further beyond the chromosome ends versus that observed for Sir2 in cells with *ysa1*-deletion, such as the right arms of chromosome 3 and 6, and the left arm of chromosome 11 (Figure 2). These findings support the notion that Sir3 is a binding partner of AAR in vitro. Furthermore, the enrichment signals of Sir3 and AAR on almost all telomeres in the *ysa1*-deletion cells were higher than those reported for wild-type cells (Figure 2). This finding supports the ability of AAR to enhance the chromatin epigenetic silencing effect. Moreover, the relative amounts of Sir proteins were not significantly different between wild type and *ysa1*-deletion cells (Appendix A). Therefore, the increased distribution and stronger association signal of Sir3 were not caused by increased concentrations of Sir3 in *ysa1*-deletion cells. These results indicate that, at some telomeres, Sir3 was able to spread along chromatin in the apparent absence of Sir2 and, perhaps, with the help of AAR binding. These data lend credence to previous findings suggesting that Sir3 overexpression resulted in its association with extended chromatin domains in the absence of Sir2 and Sir4 [33,34]. It is implied that AAR-Sir2 interaction may be merely an enzyme-product relationship and AAR-Sir3 interaction may have a more meaningful impact on true biological function.

Additionally, we used the qPCR for the validation test. The qPCR was conducted on Sir2-ChIP, Sir3-ChIP, and AAR-ChAP for 12 locations distributed on four distinct chromosomes (Figure 3). In general, all test locations were able to detect the enrichment signal through the use of actin for normalization, compared with the respective input DNA. The results were consistent with the occupancies of AAR, Sir2, and Sir3 on telomeres. 

Interestingly, the relative enrichment pattern of AAR is similar to that of Sir3 but different from that of Sir2. For instance, in both wild-type and *ysa1*-deletion cells, on the left arm of chromosome 1, Sir2 occupancy at the YAL068C gene was higher than at the YAL067C gene, and that at the YAL067C gene was higher than at the YAL065C gene (the 1st vs. the 2nd, the 2nd vs. the 3rd bar from the left, Figure 3a,e, for wild-type and *ysa1*-deletion cells, respectively). In contrast, the occupancies of both Sir3 and AAR at YAL068C gene were higher than those observed at the YAL067C and YAL065C genes (the 5th vs. the 4th and the 6th and 8th vs. the 7th and 9th bar from the left, Figure 3a,e, for wild-type and *ysa1*-deletion cells, respectively), and those at the YAL067 gene were similar to those at the YAL065C gene (the 4th vs. the 6th and the 7th vs. the 9th bar from the left, Figure 3a,e, for wild-type and *ysa1*-deletion cells, respectively). Another example can be seen. On the right arm of chromosome 6, Sir2 occupancy at 0.07 k bases from the chromosome end was higher than that at the 0.6 k region, and that at the 0.6k region was higher than that at the YFR056C gene (the 1st vs. the 2nd, the 2nd vs. the 3rd bar from the left, Figure 3c,g, for wild-type and *ysa1*-deletion cells, respectively). In contrast, the occupancies of both Sir3 and AAR at 0.6 k bases from the chromosome end were higher than those observed at the 0.07 k region and the YFR056C gene (the 5th vs. the 4th and 6th and the 8th vs. the 7th and 9th bar from the left, Figure 3c,g, for wild-type and *ysa1*-deletion cells, respectively), and those at the 0.07 k region were similar to those at the YFR056C gene (the 4th vs. the 6th and the 7th vs. the 9th bar from the left, Figure 3c,g, for wild-type and *ysa1*-deletion cells, respectively). However, we also found that the relative enrichment pattern of Sir2 on the left arm of chromosome 11 in the wild-type cells was different from that in the *ysa1*-deletion cells (the 1st vs. the 2nd vs. the 3rd bar from the left, Figure 3d,h, for wild-type and *ysa1*-deletion cells, respectively). But those of AAR and Sir3 in both wild-type and *ysa1*-deletion cells were still similar (the 4th vs. the 5th vs. the 6th and 7th vs. the 8th vs. the 9th bar from the left, Figure 3d,h, for wild-type and *ysa1*-deletion cells, respectively). 

### 3.3. Effect of AAR on Sir2 Spreading and Sir3 Spreading on Telomeres

Expanding on the concept that the binding of SIR complexes to telomeres is an indicator for the presence of silent heterochromatin, we plotted the enrichments of Sir2, Sir3, and AAR as a moving average function of distance from the telomere [35,36] in both wild-type and *ysa1*-deletion strains. We found that, for wild type cells, the enrichment ratio of the average spread of Sir3 on telomeres rapidly decreased to a basal level at a position around 15 kb from the chromosome ends. In contrast, in *ysa1*-deletion cells, the Sir3 signal only decreased gradually to basal levels beyond 22.5 kb (Figure 4a). Similar distribution patterns were also demonstrated for AAR (~15 kb vs. ~20 kb for the wild-type vs. *ysa1*-deletion cells, respectively (Figure 4b). In contrast, the distribution pattern for Sir2 in wild-type cells largely overlapped with that observed in *ysa1*-deletion cells (Figure 4c). These data imply that the increased amount of AAR in the *ysa1*-deletion cells promoted the increased distribution of Sir3 and AAR, but not Sir2, further along the telomeres than that observed in wild-type cells.

Sir3-dependent telomere clustering, a response to starvation/stress, can be attributed to the reactive oxygen species (ROS) generated by mitochondria [37]. However, the mechanism through which mitochondrial activity results in a nuclear effect is unclear. The present results may offer an explanation. The amount of AAR is increased by approximately 50% in *ysa1*-deletion cells [32], AAR is able to affect the electron transport chain and *ysa1*-deletion cells display a higher resistance to ROS [38]. The increased level of AAR may protect cells from stresses through its effects on ROS and the electron transport chain. AAR associates with Sir3, enhances the stability of Sir3 binding to chromatin, and promotes Sir3 extension further along the telomeres to form Sir3-dependent telomere clustering that may be a protective response to starvation/stress. Our results suggest that these AAR effects may induce a more obvious phenomenon of telomere clustering and be involved in the phase-separated liquid-like droplet organization of heterochromatin [39,40].

### 3.4. Effect of AAR on Sir3 Interactions In Vitro

The Sir3 N-terminal region contains a bromo-adjacent homology (BAH) domain (11-196 aa), whereas the Sir3 C-terminal region contains an AAA ATPase-like (AAL) domain (532-834 aa) and a winged helix-turn-helix domain (840-978 aa) [26,41,42]. We separated Sir3 into the Sir3N (1-214 aa) and Sir3C (215-978 aa) parts to examine the influence of AAR association on Sir3 interactions. Surprisingly, after applying an established in vitro assembly system of SIR-nucleosome pre-heterochromatin filaments [24,25], shorter rope-like structures were observed when the reaction contained only Sir3N, Sir3C, and nucleosomes (Figure 5a). In contrast, longer rope-like structures were observed when the reaction contained Sir3N, Sir3C, nucleosomes, and AAR (Figure 5b). The lengths of these rope-like structures were individually measured, and the length distributions are shown in Figure 5c. The quantification results indicated that AAR raised the length of the rope-like structure from ~70 nm to ~135 nm. However, these effects were not observed when the full-length Sir3, Sir3N, or Sir3C was used (Figure 5d–i). The reasons for these differences exhibited by the full-length Sir3 and the separated Sir3N+Sir3C should be identified. Sir2 produces AAR, which accompanies the structural rearrangement of SIR complexes [21] but the biological role of this conformational change is still unclear. We may potentially discover the key reason for the structural rearrangement of the SIR complex as induced by Sir2 activities. This conformational change may correlate with the structural or conformational alteration of the Sir3 BAH domain and C-terminal region to create a stable association space for AAR binding. It is likely that this change results in the stabilization of the intermolecular interactions between the Sir3 BAH domain and the C-terminal region through association with AAR. This renders the Sir3 dimer more stable, potentially forming a poly-Sir3 complex (denoted as polySir3-AAR). Thus, it is implied that AAR mediates a conformational change in the protein complex involved in epigenetic gene silencing.

In the absence of Sir2 and Sir4, Sir3 alone has been shown to associate with nucleosomal arrays in vitro [43]. The present results (Figure 5) support this finding. However, the majority of our purified native yeast nucleosomes were monomers, dimers, and tetramers. Thus, for longer SIR-nucleosome filaments, protein–protein interactions must occur. Furthermore, multiple interactions must link shorter multimeric nucleosomes to create longer polymeric nucleosomes and, ultimately, long SIR-nucleosome pre-heterochromatin filaments.

## 4. Conclusions

The ability of AAR to modulate the formation of the telomere silent heterochromatin regions that may not be totally mediated by Sir2. Another non-Sir2 AAR interaction protein, such as Sir3, may also form polySir3-AAR in silent heterochromatin regions. As illustrated in the proposed model (Figure 6), AAR plays a critical role in stabilizing Sir3 interactions on the telomere silent heterochromatin domain, and AAR remains associated with Sir3 (in the apparent absence of Sir2 and Sir4) in the extended silent heterochromatic region.

## Figures and Tables

**Figure 1 genes-10-00577-f001:**
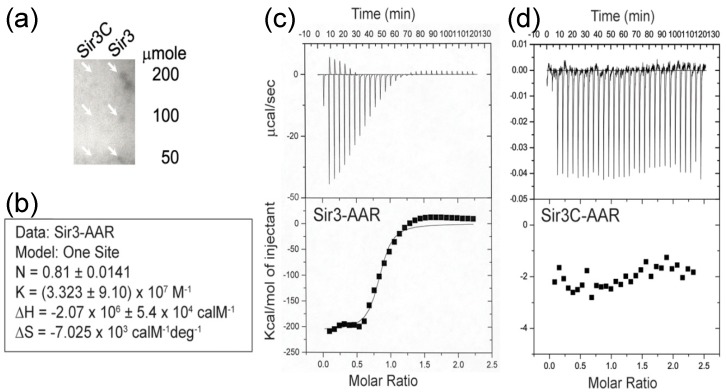
Interaction of O-acetyl-ADP-ribose (AAR) with silent information regulator (Sir)3. (**a**) The dot blotting assay showed the binding of ^32^P-AAR to the indicated amount of Sir3 and Sir3C. The white arrows point out the detected individual signal on the dot positions of Sir3 but no signal on the dot positions of Sir3C. (**b**) Summary of the data analysis in the isothermal titration calorimetry (ITC) assay of Sir3-AAR using the MicroCal Origin software package, assuming one set of sites to obtain the binding constant (K), reaction Stoichiometry (N), enthalpy (ΔH), and entropy (ΔS) with their standard error. (**c**–**d**) The raw thermograms of ITC data were obtained after each injection (top panels). The integrated curves comprising experimental data points (filled squares) and best fit (curved line) with the one site model are displayed (bottom panels). The interaction between AAR and Sir3 was observed (**c**); however, there was no interaction observed between AAR and Sir3C (**d**).

**Figure 2 genes-10-00577-f002:**
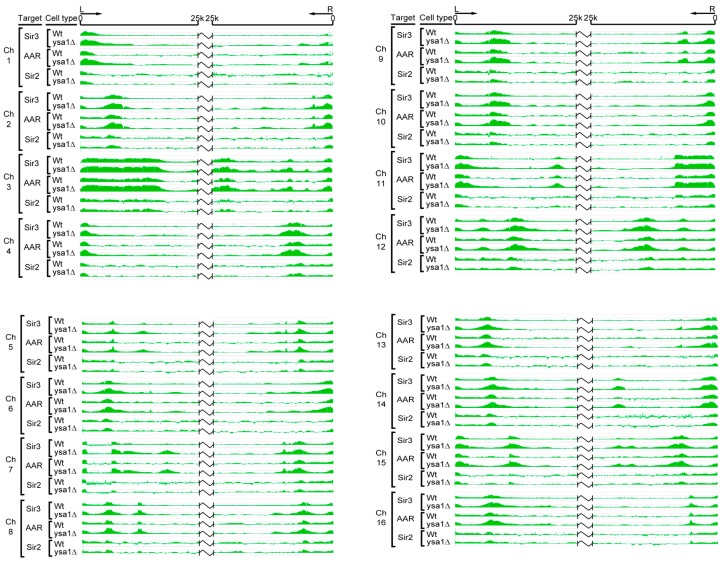
Chromosomal distribution of Sir2, Sir3, and AAR over telomeric regions. The chromosomal distributions of Sir3, Sir2, and AAR were determined through chromatin immunoprecipitation (ChIP)-on-chip and chromatin affinity purification (ChAP)-on-chip assays. For each chromosome, data are presented for a 25 kb region from the left (L) and right (R) of the chromosome ends for Sir3 in wild-type cells, Sir3 in *ysa1*-deletion cells, AAR in wild-type cells, AAR in *ysa1*-deletion cells, Sir2 in wild-type cells, and Sir2 in *ysa1*-deletion cells, as indicated.

**Figure 3 genes-10-00577-f003:**
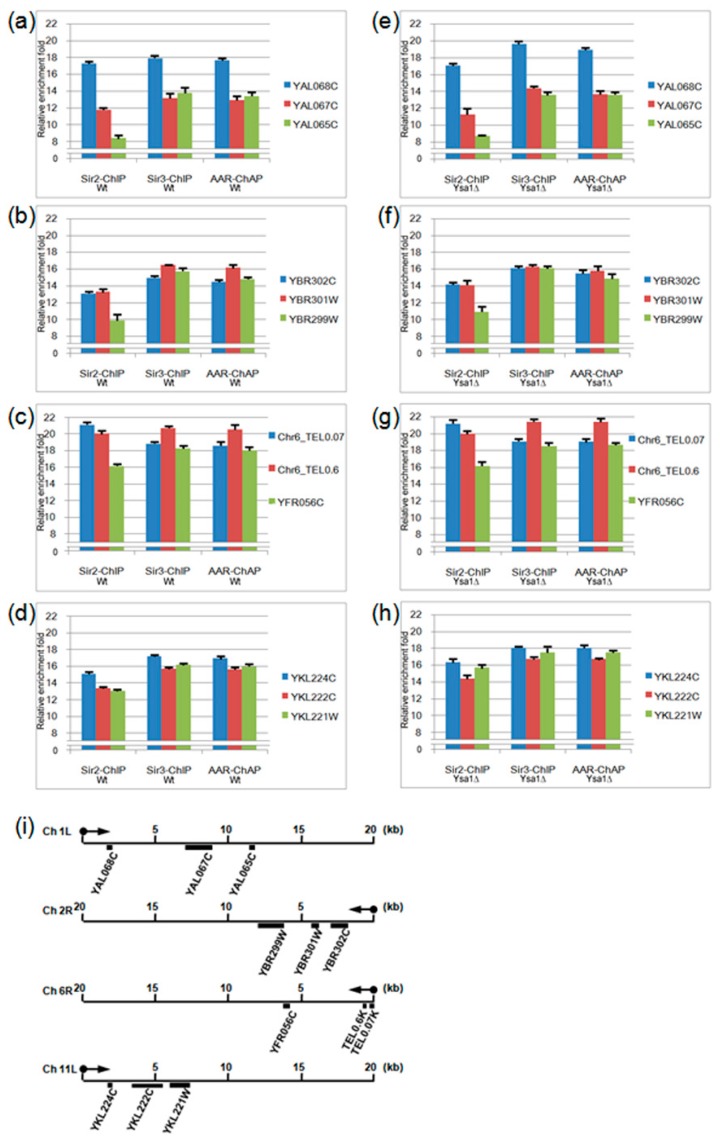
The quantitative polymerase chain reaction (qPCR) results of Sir2-ChIP, Sir3-ChIP, and AAR-ChAP. (**a**–**h**). The relative fold-change enrichment of the Sir2, Sir3, and AAR occupancy signals were compared with the respective input DNA. Three test genes/locations on the left arm of the chromosome 1 telomere (**a**,**e**), right arm of chromosome 2 telomere (**b**,**f**), right arm of chromosome 6 telomere (**c**,**g**), or left arm of chromosome 11 telomere (**d**,**h**) for wild type (**a**–**d**) or *ysa1*-deletion (**e**–**h**) cells are indicated, respectively. (**i**) Illustration indicating the individual relative location of each gene on different telomeric regions. Cell type: Wt: wild-type strain; ysa1Δ: *ysa1*-deletion strain.

**Figure 4 genes-10-00577-f004:**
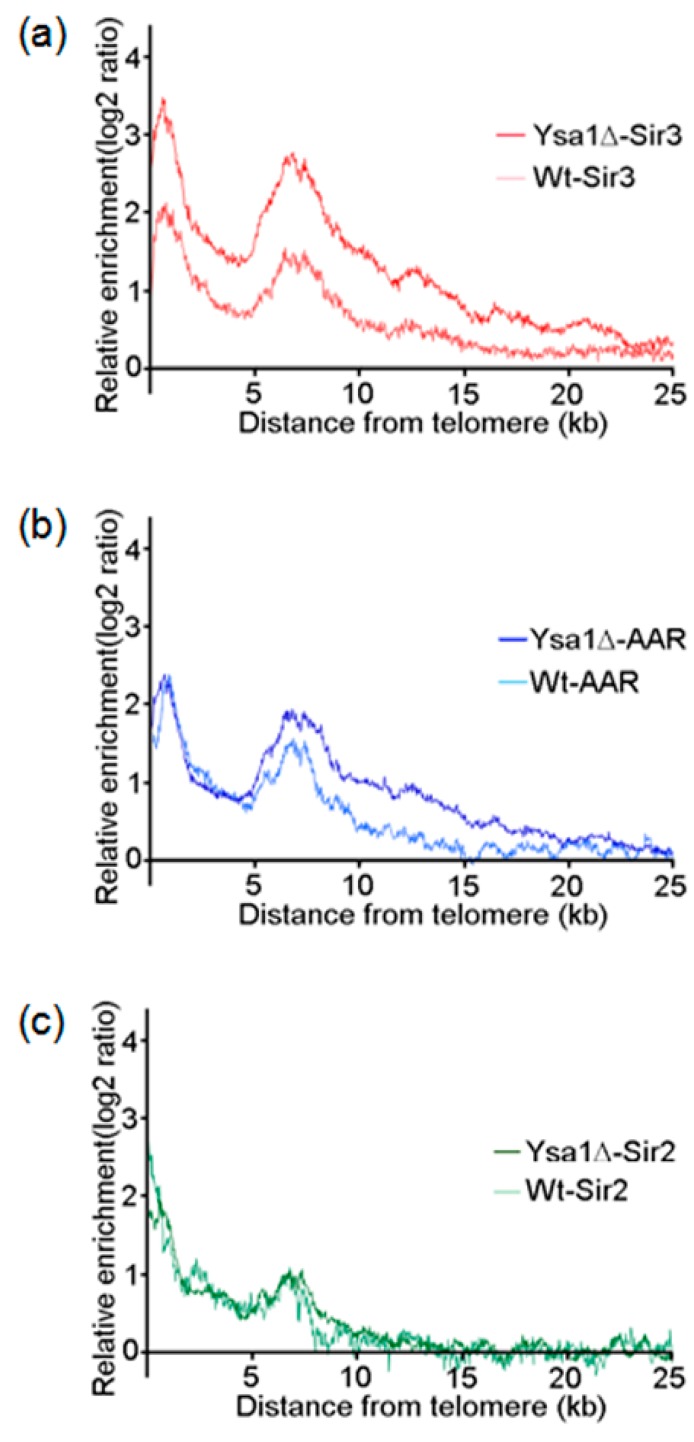
Occupancies of AAR, Sir2, and Sir3 on telomeres. (**a**–**c**) The moving averages of Sir3 (**a**), AAR (**b**), and Sir2 (**c**) binding for all 32 yeast telomeres are plotted as a function of distance from the chromosome ends. Binding enhancement was measured as the log2 score of immunoprecipitation versus input. Data are provided for Sir3, AAR, and Sir2 binding in either wild type or *ysa1*-deletion cells, as indicated. Cell type: Wt: wild-type strain; ysa1Δ: *ysa1*-deletion strain.

**Figure 5 genes-10-00577-f005:**
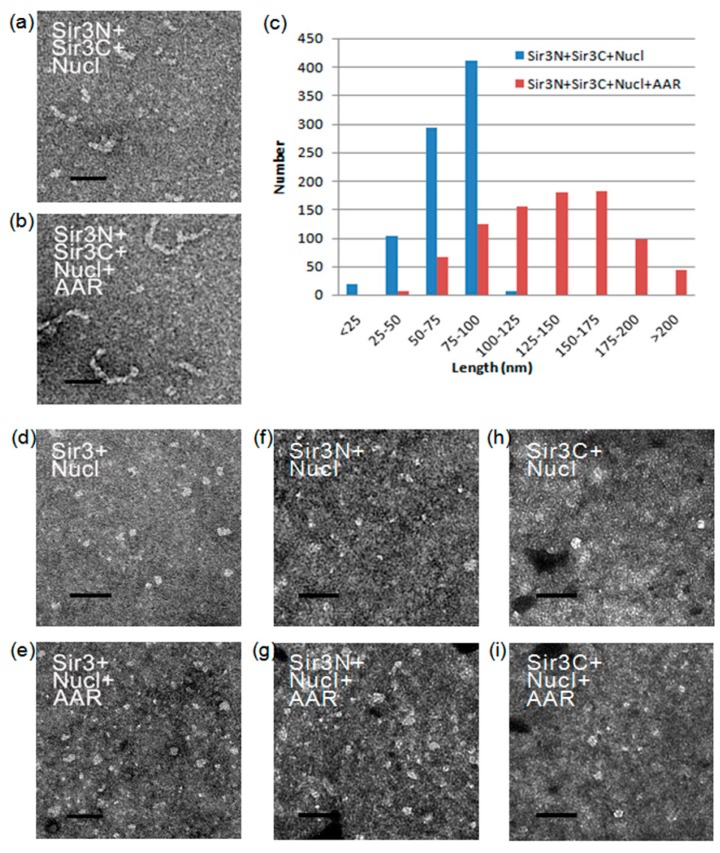
Morphology of Sir3-AAR-nucleosome particles. (**a–b**) Electron micrographs show the results of the assembly reactions containing yeast nucleosomes, Sir3N, and Sir3C in the absence (**a**) or presence (**b**) of AAR, respectively. (**c**) The length distributions of Sir3NC-nucleosome and Sir3NC-AAR-nucleosome are displayed. The quantification of each EM examination was performed in at least triplicate. The length of the particles was measured in randomly pick-up from at least three distinct fields of view on the electron microscopy (EM) grid. (**d**–**i**) Electron micrographs show the results of the assembly reactions containing yeast nucleosomes and Sir3 (**d**–**e**) (or Sir3N (**f**–**g**) or Sir3C (**h**–**i**)) in the absence (**d,f,h**) or presence (**e,g,i**) of AAR, respectively. The scale bar represents 100 nm.

**Figure 6 genes-10-00577-f006:**
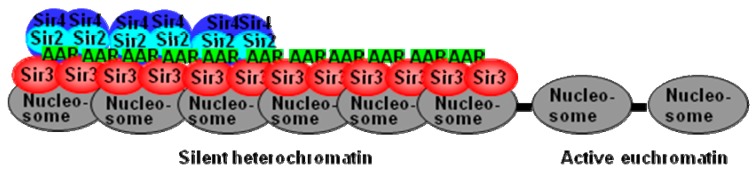
The proposed model of polySir3-AAR on telomeres. On the initial establishment step of silent heterochromatin formation, Sir2 associates with Sir3 and Sir4 as an SIR complex, and then a repeated recruitment cycles of SIR complexes along the adjacent chromatin region to form a silent heterochromatin domain. In the extended silent heterochromatic region, additional maturation processes like AAR continue to be associated with Sir3, in the absence of Sir2 and Sir4, which stabilize and maintain the silent heterochromatin.

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
