# Peer review of "The Capability of O-Acetyl-ADP-Ribose, an Epigenetic Metabolic Small Molecule, on Promoting the Further Spreading of Sir3 along the Telomeric Chromatin"

_genes, 2019, doi:10.3390/genes10080577_

Round 1
Reviewer 1 Report
The authors have addressed many of the scientific limitations of the manuscript in this first revision. The Sir3-biochemistry in the new figure 1 helps, particularly the ITC. The quantitation of the images in the new figure 4 are welcomed. However, additional revision is required for the new figure 3d, and new figure 4 requires some additional controls, as stated below. The improvement in the English was appreciated but there are still issues that must be rectified, as itemized below.
Issues with the science:
Figure 3d – The authors claim that the ChIP-qPCR in the figure matches the composite ChIP-on-ChIP data in figures 3a-c. However, there is too much data in figure 3d to easily reach this conclusion. Moreover, the major point of figures 3a-c is that Sir3 and AAR are more enriched at proximal locations in the ysa1 mutant, and that both spread farther from the chromosome end in the ysa1 mutant. The authors don’t discuss this for figure 3d, and it is hard to tell from the data that this is really the case. This paper is supposed to be about the impact of AAR on Sir3! Unfortunately, the only thing the authors describe is that Sir3 spreads farther than Sir2 at Tel6R, and that AAR has no impact on Sir2. Thus, figure 3d does not really recapitulate the important data of figures 3a-c.
The error bars are suspiciously small and consistent from sample to sample. The authors need to define exactly what the error bars represent (which trials were used to calculate error and how was the error calculated). This is critical because the magnitudes of the differences they report are very, very small. For example, Sir2 levels drop only ~5% at the 0.6 kb site relative to the chromosome end. Similarly, Sir3 levels increase only ~10% at the 0.6 kb site relative to the chromosome end. Do the measurements of error validate a discussion of 5-10% differences by ChIP-qPCR?
Since only one of four telomeres is discussed (Tel6R), I recommend that only that only Tel6R be shown in the figure and the rest of the data be moved to the supplement.
Figure 4 – Can long ropes be formed with just Sir3C, nucleosomes and AAR? Can they be formed with just Sir3N, nucleosomes and AAR? Are nucleosomes even required for these ropes? These controls are easy and need to be included. It is likely that the images already exist.
Issues with English:
Lines 3 and 85 – what does “relative” mean?
Lines 86, 91, 97 – need corrections.
Line 184 – “expression” levels?
Line 314 – needs corrections.
Lines 362-363, 364-365 and 373-376 are redundant.
Lines 455-464 – It is still not at all clear what the authors are trying to say. Does a ysa1 null influence telomere clustering?
Lines 478-485 – The speculation described here is difficult to follow. Too confusing.
Lines 484-485 – How does stabilizing a Sir3 dimer lead to polymers? Are the authors suggesting that AAR strengthens dimers of dimers? Are the authors suggesting that there are intermolecular interactions between the Sir3N domain of one Sir3 with the Sir3C of another Sir3? This section is too confusing.
Line 723 – Protein-protein interactions? Which proteins?
Lines 727-731 – These lines do not describe the figure. If the paragraph is supposed to be a conclusion, it does not read like it.
Issues with figures:
Figure 1 - The panels are not aligned with one another
Figure 5 – My version of this figure is still distorted. Something must be causing the distortions in preparation of the reviewer's copy.
Author Response
Comments and Suggestions for Authors
The authors have addressed many of the scientific limitations of the manuscript in this first revision. The Sir3-biochemistry in the new figure 1 helps, particularly the ITC. The quantitation of the images in the new figure 4 are welcomed. However, additional revision is required for the new figure 3d, and new figure 4 requires some additional controls, as stated below. The improvement in the English was appreciated but there are still issues that must be rectified, as itemized below.
We appreciate reviewer's recognition on our revised manuscript. We have now added some more figures and the corresponding description in our new revise version of manuscript. We have now also reedited some other paragraphs regarding to reviewer's commends.
Issues with the science:
Figure 3d – The authors claim that the ChIP-qPCR in the figure matches the composite ChIP-on-ChIP data in figures 3a-c. However, there is too much data in figure 3d to easily reach this conclusion. Moreover, the major point of figures 3a-c is that Sir3 and AAR are more enriched at proximal locations in the ysa1 mutant, and that both spread farther from the chromosome end in the ysa1 mutant. The authors don’t discuss this for figure 3d, and it is hard to tell from the data that this is really the case. This paper is supposed to be about the impact of AAR on Sir3! Unfortunately, the only thing the authors describe is that Sir3 spreads farther than Sir2 at Tel6R, and that AAR has no impact on Sir2. Thus, figure 3d does not really recapitulate the important data of figures 3a-c.
We thank for reviewer's criticism. We have now reedited the Figure 3d-3e as the new Figure 3a-3i and the Figure 3a-3c as the new Figure 4a-4c. We have also added some paragraphs of the corresponding description (lines 196-200, page 6 & lines 208-230, page 8). Yes, the reviewer's comment is right. We addressed the issue regarding to the impact of AAR on Sir3 and we just focused on the AAR promotes the Sir3 spreading further along the telomeric region in our current present story.
The error bars are suspiciously small and consistent from sample to sample. The authors need to define exactly what the error bars represent (which trials were used to calculate error and how was the error calculated). This is critical because the magnitudes of the differences they report are very, very small. For example, Sir2 levels drop only ~5% at the 0.6 kb site relative to the chromosome end. Similarly, Sir3 levels increase only ~10% at the 0.6 kb site relative to the chromosome end. Do the measurements of error validate a discussion of 5-10% differences by ChIP-qPCR?
We thank to reviewer mentions this. The error bars were generated from 1 standard deviation as the common statistical used. In fact, we was a little surprised to the data of standard deviation in the qPCR test. We calculated the averages and their standard deviations using the classical formula/function in the microsoft office excel software. The data of standard deviation are shown as below:
Ct-SD | ||||||
Wt | Wt | Wt | Dysa1 | Dysa1 | Dysa1 | |
Sir2-ChIP | Sir3-ChIP | AAR-ChAP | Sir2-ChIP | Sir3-ChIP | AAR-ChAP | |
Actin | 0.14 | 0.11 | 0.17 | 0.17 | 0.04 | 0.20 |
YAL068C | 0.03 | 0.04 | 0.03 | 0.08 | 0.08 | 0.04 |
YAL067C | 0.03 | 0.11 | 0.10 | 0.35 | 0.02 | 0.12 |
YAL065C | 0.05 | 0.37 | 0.09 | 0.02 | 0.10 | 0.08 |
YBR302C | 0.02 | 0.04 | 0.03 | 0.04 | 0.03 | 0.12 |
YBR301W | 0.06 | 0.00 | 0.04 | 0.14 | 0.03 | 0.20 |
YBR299W | 0.57 | 0.09 | 0.02 | 0.20 | 0.04 | 0.19 |
Chr6_TEL0.07 | 0.11 | 0.03 | 0.16 | 0.12 | 0.06 | 0.07 |
Chr6_TEL0.6 | 0.14 | 0.02 | 0.20 | 0.08 | 0.07 | 0.11 |
YFR056C | 0.07 | 0.08 | 0.14 | 0.12 | 0.10 | 0.04 |
YKL224C | 0.10 | 0.05 | 0.10 | 0.11 | 0.01 | 0.11 |
YKL222C | 0.04 | 0.05 | 0.07 | 0.15 | 0.04 | 0.01 |
YKL221W | 0.05 | 0.05 | 0.07 | 0.07 | 0.28 | 0.06 |
Since only one of four telomeres is discussed (Tel6R), I recommend that only that only Tel6R be shown in the figure and the rest of the data be moved to the supplement.
We thank for reviewer's suggestion. We have now reedited the Figure 3d-3e as the new Figure 3a-3i and the Figure 3a-3c as the new Figure 4a-4c. We have also added some paragraphs of the corresponding description (lines 196-200, page 6 & lines 208-230, page 8).
Figure 4 – Can long ropes be formed with just Sir3C, nucleosomes and AAR? Can they be formed with just Sir3N, nucleosomes and AAR? Are nucleosomes even required for these ropes? These controls are easy and need to be included. It is likely that the images already exist.
We thank for reviewer's criticism. We have now added the Sir3N+Nucleosomes and Sir3C+Nucleosomes in absent or present AAR as other control tests (new Figure 5f-5i) and reedited the corresponding text as "....these effects were not observed when the full-length Sir3, Sir3N or Sir3C was used (Figure 5d-5i)...." (line274-275, page 10). And since the AAR does not interact with nucleosome and Nucleosome+AAR can not form the filament structure (Onishi et al., 2007; Tung et al., 2017), so we did not re-examine and add the control of Nucleosome+AAR in this paragraph.
Issues with English:
Lines 3 and 85 – what does “relative” mean?
We thank for reviewer's criticism. AAR is generated by Sir2 during its action on a histone epigenetic modification. That is, the production of AAR is an epigenetic action-dependent/relative event. So, originally, we call AAR is an epigenetic metabolic small molecule. Then after another reviewer's criticism, we consulted with an English science editor, so we change to call AAR as an epigenetic relative metabolic small molecule. In fact, both "an epigenetic relative small molecule" and "an epigenetic relative metabolic small molecule" should be acceptable. And we prefer to call AAR as "an epigenetic metabolic small molecule". So, we now correct back to "AAR is an epigenetic metabolic small molecule".
Lines 86, 91, 97 – need corrections.
We thank to reviewer points out this. We have now reedited them as ".... O-acetyl-ADP-ribose (AAR), an epigenetic metabolic small molecule, is generated by NAD-dependent Sir2/Class III histone deacetylase...." (lines 46-48, page 2), ".... assembly system of SIR-nucleosome pre-heterochromatin filaments that has been established. All purified yeast nucleosomes, Sir2, Sir3, Sir4, and AAR are required for this assembly, and the formation of these filaments is modulated...." (lines 52-54, page 2), and ".... AAA ATPase-like domain in the Sir3 C-terminal region that may interact with ATP [26], and AAR exhibits the structural similarity to ATP...." (lines 57-59, page 2).
Line 184 – “expression” levels?
We thank to reviewer points out this. We have now re-written it as ".... expressional levels were...." (lines 116-117, page 3).
Line 314 – needs corrections.
We thank to reviewer points out this. We have now re-written it as ".... arrows were pointed out the detected individual signal on the dot positions of Sir3 but no signal on the dot positions...." (lines 138-139, page 4).
Lines 362-363, 364-365 and 373-376 are redundant.
We thank to reviewer mentions this. To avoid the redundancy, we have now deleted the sentence: "Moreover, both Sir3 and AAR were colocalized over regions that clearly extended further beyond the chromosome ends compared with Sir2.".
Lines 455-464 – It is still not at all clear what the authors are trying to say. Does a ysa1 null influence telomere clustering?
We thank for reviewer's criticism. In this paragraph, we discussed that the mechanism through the mitochondrial activity results in a nuclear effect of telomere cluster and tried to make an explanation for that. Sir3-dependent telomere clustering can be attributed to the mitochondrial activity of reactive oxygen species (ROS). AAR affects the electron transport chain and ysa1-deletion cells display higher resistance to exogenous ROS. Our present results may offer a potential answer. The amount of AAR is increased by approximately 50% in ysa1-deletion cells. And the increased level of AAR may protect cells from stress through its effects on ROS and the electron transport chain. AAR associates with Sir3 and increases their further spreading along the silent heterochromatin, thus, to form Sir3-dependent telomere clustering that may be a protective response to starvation/stress. In summary, these AAR effects may induce a more obvious phenomenon of telomere clustering and be involved in the phase-separated liquid-like droplet organization of heterochromatin.
Lines 478-485 – The speculation described here is difficult to follow. Too confusing.
We thank for reviewer's criticism. In this part, firstly, we described an open question: "Sir2 produces AAR, which accompanies the structural rearrangement of SIR complexes but the biological role of this conformational change is still unclearly". Then we discussed a potentially answer according to our data. So, we proposed that this conformational change may correlate with the structural or conformational alteration of the Sir3 BAH domain and C-terminal region to create a stable association space for AAR binding. It is likely that this results in stabilization of the intermolecular interactions between the Sir3 BAH domain and the C-terminal region through association with AAR.
Lines 484-485 – How does stabilizing a Sir3 dimer lead to polymers? Are the authors suggesting that AAR strengthens dimers of dimers? Are the authors suggesting that there are intermolecular interactions between the Sir3N domain of one Sir3 with the Sir3C of another Sir3? This section is too confusing.
Yes, reviewer is right. We proposed that the stabilization of intermolecular interactions between the Sir3 BAH domain and the C-terminal region through association with AAR. We are now working to solve the high resolution of the Sir3-AAR structure via cryo-EM. We believe it is beyond the scope of the present story. However, we have a model as below:
Line 723 – Protein-protein interactions? Which proteins?
We thank reviewer points out this. The "protein-protein interactions" means any kind interaction that is able to promote the longer SIR-nucleosome filament formation, such as, Sir3-Sir3 interaction and Sir3-nucleosome interaction, however, other potential interactions (ex: nucleosome-nucleosome interaction) cannot be totally ruled out.
Lines 727-731 – These lines do not describe the figure. If the paragraph is supposed to be a conclusion, it does not read like it.
We thank for reviewer's criticism. We have now reedited this paragraph as "The ability of AAR for modulating the formation of the telomere silent heterochromatin regions that may be not totally mediated by Sir2. Another non-Sir2 AAR interaction protein, such as Sir3 may also form polySir3-AAR in the silent heterochromatin regions. As illustrated in the proposed model (Figure 6), AAR plays a critical role in stabilizing Sir3 interactions on the telomere silent heterochromatin domain, and AAR remains associated with Sir3 (in the apparent absence of Sir2 and Sir4) in the extended silent heterochromatic region" (lines 302-307, page11).
Issues with figures:
Figure 1 - The panels are not aligned with one another
The figure 1 looks OK, but figure 2 (on for reviewer version made by journal/ publisher) looks something wrong. Here is the Figure 1 & 2 as below:
Figure 5 – My version of this figure is still distorted. Something must be causing the distortions in preparation of the reviewer's copy.
Here is the figure 6 (figure 5 of old version):

Reviewer 2 Report
The manuscript has significantly approved and is now suitable for publication.
Author Response
We appreciate reviewer's recognition on our revised manuscript
Round 2
Reviewer 1 Report
The authors have addressed my concerns. The English is still clumsy. They should consider finding a new consultant to evaluate their scientific English.
This manuscript is a resubmission of an earlier submission. The following is a list of the peer review reports and author responses from that submission.
Round 1
Reviewer 1 Report
AAR is a metabolic byproduct of the deacetylation reaction carried out by Sir2. The Liou lab showed previously that AAR enhances silencing by the Sir2/3/4 complex, possibly through interactions with Sir2. In this manuscript, Tung et al. provide preliminary evidence that AAR might act through Sir3. The authors compare the chromosomal distribution of Sir2 and Sir3 by ChIP to the distribution of chromatin fragments that can be precipitated by immobilized AAR (the ChAP technique). In addition to precipitating regions bound by both Sir2 and Sir3, AAR also precipitates regions bound solely by Sir3. Moreover, when AAR levels are increased by eliminating an AAR metabolizer (YSA1), the Sir3 bound domains increase in size and density. The Sir2 domains do not. Together, the data suggest that AAR favors Sir3 binding. Second, the authors analyze interactions between purified nucleosomes and Sir3 by EM. Curiously, N and C-terminal fragments of Sir3 form rope-shaped particles with nucleosomes only when AAR is added. This suggests that the AAR acts directly on Sir3 or nucleosomes. These data, with some additional biochemistry, would greatly interest the field.
Major issues:
1) This paper must be heavily edited by someone with greater command of English. It is very hard to read as is.
2) The paper would be greatly strengthened by evidence that Sir3 and AAR interact directly. It is not sufficient to cite unpublished data.
3) Figure 2a – The authors indicate that Sir3 binding in a wt strain drops to background levels by 15kb from a telomere. How is background defined?
4) Figure 2d - Error bars are required, as well as an indication that sufficient trials were performed. Currently, it is not clear whether any of the differences reported are significant. The figure is confusing and too dense with data to be easily interpretable.
5) Lines 170-171 – Sir3 spreads on some telomeres without Sir2, and the authors suggest that AAR helps this spreading. If so, then overexpression of YSA1 should reduce these Sir2-free domains of Sir3. This experiment should be easy, given the data presented in Wang and Liou, 2019. The predicted result would strengthen the idea that elevated AAR levels facilitate Sir3 binding rather than simply correlate with Sir3 binding.
6) Figure 3 - There is no quantitation of images. Distribution of rope length should be reported, as well as frequency of appearance. It is not sufficient to show a single representative image.
7) Figure 4 – The figure has no labels. What do the green and red objects represent? The labels of the blue objects are obscured. Perhaps the figure was distorted during preparation of the reviewer’s copy.
Minor issues
1) Line 110, 125 – ChAP does not map where AAR “is” on telomeres. It maps where AAR “can bind” to telomeres in a ChAP reaction in vitro. The difference in word choice is significant.
2) Lines 130-131 appear to be redundant with the other sentences in the paragraph.
3) Line 166-168. The text is confusing because it returns to discussing figure 1 after having finished figure 2. These observations should be discussed immediately after line 112.
4) Lines 176-183 – extremely confusing and doesn’t help the paper.
5) lines 198-201 and lines 212-216. It is not clear what the authors are trying to say.
Reviewer 2 Report
A capability of O-acetyl-ADP-ribose, an epigenetic metabolic small molecule, on promoting the further spreading of Sir3 along the telomeric chromatin is not a final manuscript, author should work more on the layout of manuscripts and data presentations.
There are grammar mistakes, meaning sentences without sense, from Abstract to Conclusions.
Authors should use word epigenetic more carefully. I do not think to call AAR an epigenetic metabolic small molecule is right as it has no meaning.
The introduction is not comprehensive and detailed for non-experts in the SIR's field.
The abstract needs to be rewritten and redefined.
Authors conclusion chapter is more discussion than conclusions as they do not prove that there is direct interaction between Sir3 and AAR.Why authors do not provide data for this direct interaction? It leaves unnecessary space for doubts and undermines proposed mechanism which is based on direct interaction while this is not supported with data.
Line 19. Sir3 is also NAD-dependent histone deacetylase and the main player in this manuscript. Sir3 should be used as an example, not Sir2.
Line 22. I do not understand what point authors want to make with this sentence
Line 20 and 23. Is duplication of information. Sentences should be merged to provide coherent information.
Methods:
Line 75. It is impossible to repeat ChIP-on-chip and ChAP-on-chip assays based on the provided description. Crucial information like used amount of antibodies and AAR is needed to assess the results. What controls authors used?
Overall more information is needed in the methods chapter.
Results
112-137
Linked with methods line 75.
Importantly in these crucial experiments on which authors based the whole story, they did not use negative control such as isotype match control to confirm the signal they see is real. Also, to control AAR binding other molecules such ATP should be used as controls.
Line 142-144. I do not agree with the conclusion made here. There is no way to conclude that one rapidly decreases and the other one gradually decreases from these plots. There is fold change, yes, but the profile looks similar.
Fig.2a-c. It would be also informative to see ysa1del-Sir3 with plotted AAR profile and vice versa, whether profiles overlap of Sir3 and AAR in the same genotype.
Fig 2d. How many times was this experiment performed? There are no statistics, error bars.
Figure 4. This figure needs to be better. It’s hard to read, it does not clearly explain the authors’ proposed model. The legend should be more detailed.
